# Potential of methanol extracts of *Nephelium lappaceum* (Sapindaceae) and *Hyphaene thebaica* (Arecaceae) as adjuvants to enhance the efficacy of antibiotics against critical class priority bacteria

**Armel Jackson Seukep**[1]*, **Fula Mabel Tamambang**[1], **Valaire Yemene Matieta**[2], **Helene Gueaba Mbuntcha**[2], **Francis Desire Tatsinkou Bomba**[1], **Victor Kuete**[2], **Lucy M. Ayamba Ndip**[3,4]*

1 Department of Biomedical Sciences, Faculty of Health Sciences, University of Buea, Buea, Southwest Region, Cameroon, 2 Department of Biochemistry, Faculty of Science, University of Dschang, Dschang, West Region, Cameroon, 3 Department of Microbiology and Parasitology, Faculty of Science, University of Buea, Buea, Southwest Region, Cameroon, 4 Laboratory for Emerging Infectious Diseases, University of Buea, Buea, Southwest Region, Cameroon

* seukep.armel@ubuea.cm (AJS); lndip@yahoo.com (LMAN)

**Data Availability Statement:** All relevant data are within the manuscript and its Supporting

## Abstract

Botanicals have shown promise in mitigating antibiotic resistance in bacteria. This study examined the efficacy of methanolic extracts from two food plants (*Nephelium lappaceum* and *Hyphaene thebaica*), alone and in combination with antibiotics, against critical class priority bacteria, including multi-drug resistant (MDR) strains and clinical isolates of *Staphylococcus aureus*, *Klebsiella pneumoniae*, *Pseudomonas aeruginosa*, *Enterobacter aerogenes*, and *Escherichia coli*. The herbals underwent testing using a 96-well microplate serial dilution technique before being analyzed for their effects on bacterial cell membrane integrity and $H^+$-ATPase-mediated proton pumping. Phytochemical analysis was carried out using established techniques. The bioactive extracts displayed very good to weak antibacterial activities ($128 \leq$ MIC $\leq 2048$ µg/mL). The bark, leaf, and peel extracts of *N. lappaceum* were found to be effective against all studied MDR bacteria. *N. lappaceum* leaf extract exhibited the best activity ($128 \leq$ MIC $\leq 1024$ µg/mL on all studied MDR bacteria). Interestingly, all MBC/MIC ratios calculated were $\leq 4$, suggesting bactericidal activities. *N. lappaceum* leaf extract has shown significant inhibition of bacterial $H^+$-ATPase-mediated proton pumping and changes in the cell membrane integrity, suggesting possible modes of action. *N. lappaceum* (leaves and peels) and *H. thebaica* (fruits) extracts demonstrated a notable potential to synergize with tetracycline, vancomycin, imipenem, ciprofloxacin, and cefixime (up to 8-fold reduction of the antibiotic's MIC was recorded). *N. lappaceum* leaves and peels, and fruits of *H. thebaica* significantly improved the efficacy of all antibiotics tested against *K. pneumoniae* ATCC11296 at MIC/2. Similar effects were observed against *P. aeruginosa* PA01 and *E. coli* AG100, respectively, with leaves and peels of *N. lappaceum*. No antagonistic interactions were recorded. Qualitative phytochemical screening revealed the

Information files. We confirm that our submission contains all raw data required to replicate the results of the study. Moreover, there are no ethical or legal restrictions on sharing data.

**Funding:** The author(s) received no specific funding for this work.

**Competing interests:** The authors have declared that no competing interests exist.

presence of tannins, phenols, and saponins in all test extracts. The findings of this study are promising and suggest that *N. lappaceum* and *H. thebaica* can be used either for direct action on bacteria or to revitalize outdated antibiotics that are gradually losing their potency due to resistance.

## Introduction

The current arsenal of antibiotics is becoming less effective as pathogens are evolving and developing strategies to resist their effects. According to 2019 data on the burden of infectious diseases, drug-resistant infections caused approximately 5 million deaths, out of which 1.2 million were solely attributed to drug-resistant bacteria [1]. If the issue is not addressed promptly, the number of deaths caused by it could reach 10 million per year by 2050. This could significantly increase global healthcare costs, ranging from $300 billion to over $1 trillion per year by 2050 [1]. Antimicrobial resistance (AMR) is a major global public health threat, with few compelling antibacterial discoveries [2]. As a result, there is an increased demand for novel antibiotic alternatives or chemicals capable of re-sensitizing resistant microorganisms to traditional treatments. Because of their high antibiotic resistance profile, the ESKAPEE group (*Enterococcus faecium*, *Staphylococcus aureus*, *Klebsiella pneumoniae*, *Acinetobacter baumannii*, *Pseudomonas aeruginosa*, *Enterobacter spp*., and *Escherichia coli*) of bacteria are given special attention, and they are used to guide research and discovery of innovative antibacterial drugs [3]. Several measures are being taken to combat multi-drug resistance (MDR) in bacteria [4]. Recent decades have seen a surge in interest in medicinal plants, including edible plants and their derived phytoconstituents, where research has revealed immense potential in the battle against resistant pathogens, either alone or in combination with conventional treatments [5, 6]. The African flora, particularly that of Cameroon, shows a great diversity of medicinal plants with a wide range of biological activities, including antimicrobial properties [7, 8]. A recent book chapter provides an appraisal of well-established antibacterial agents from plants, reporting on over a hundred antibacterial phytochemicals. These compounds have shown particular effectiveness against MDR strains, with some already in clinical use and others in various stages of development [6]. Interestingly, there have been few to no reports of resistance against antibacterial phytochemicals so far [9]. Another promising solution to this problem involves using a combinational approach that utilizes the interaction between plant extracts and conventional antibiotics. This method is considered the most effective way to combat this issue. Combining plant extracts with antibiotics can enhance the overall antimicrobial effect, and the extracts can be used as resistance-modifying or modulating agents [10]. This approach can potentially be a game-changer in the fight against antibacterial resistance. Studies have shown that plant-derived compounds can enhance the effectiveness of conventional antibiotics by working synergistically with them. Several key examples of the synergistic activity of antibiotics and bioactive plant extracts were previously documented [11–13]. Using plant extracts with traditional medication is widespread, particularly in Cameroon. The findings show that herbal remedies, particularly edible plant extracts [14], can significantly improve the efficiency of standard antibiotics against MDR bacteria, as demonstrated by several studies in Cameroon [15–22]. As a result, the capacity of plant chemicals to repurpose traditional antibiotics has the potential to significantly improve world health by battling drug-resistant pathogens.

The antibacterial and combination efficacy of *Nephelium lappaceum* (Sapindaceae) and *Hypheane thebaica* (Arecaceae) extracts with conventional medicines have not been widely

studied on MDR strains. *N. lappaceum* and *H. thebaica* are two plants commonly consumed in Cameroon, and reports also show their use in traditional medicine in the management of several ailments, including infectious diseases. *N. lappaceum*, also called Rambutan, is reported in folk medicine to treat diabetes and hypertension. Its fruits are used as anthelmintic, antidiarrheal, and against dysentery, whereas the leaves are used as poultices to manage headaches [23]. The antioxidant and antibacterial activities of the peel and seed extracts of *N. lappaceum* were reported [23]. Various solvent extracts (ethanol, water) of *N. lappaceum* depicted antibacterial activity against *Staphylococcus aureus*, *Streptococcus pyogenes* [24], and MDR *Pseudomonas aeruginosa* [25]. A recent study by Florenly et al. [26] displayed the antibacterial efficacy of the plant nanoparticles against oral bacteria, including *S. aureus* and *Streptococcus mutans*. *N. lappaceum* contains important bioactive components (saponins, tannins, flavonoids, steroids, cardiac glycosides, phenols, and alkaloids), which are known to be the contributing factors to its pharmacological properties [23, 26]. *H. thebaica*, also known as the Doum plant, is traditionally used for firewood and charcoal; the leaves are used in making mats, brooms, ropes, textiles, strings, and basketry. The roots are used to treat bilharzia diseases, and fruits are chewed to control hypertension [27]. Pharmacologically, *H. thebaica* is well-known for its antioxidant, anticancer, and anti-inflammatory potential. Its phenolic and flavonoid content has been explored for antimicrobial potential against various Gram-positive and Gram-negative bacteria and fungal pathogens. *H. thebaica* contains bioactive compounds comprising phenol, flavonoids, cardiac glycosides, tannins, and saponins, all of which contribute to its pharmacological properties [27]. The current study investigated the antibacterial properties of *N. lappaceum* and *H. thebaica* methanol extracts and their ability to improve the efficacy of conventional antibiotics against some bacteria of the ESKAPEE group. The modes of action of the most active extract were also examined as well as the qualitative phytochemical analysis.

## Materials and methods

### Plant material and extraction

*Nephelium lappaceum* (peels, bark, leaves, and seeds) was sourced from Penja (Penja fruit market) in the Littoral Region of Cameroon, while *Hyphaene thebaica* (fruits) was obtained from Yagoua (Yagoua market, Far North Region, Cameroon), in August 2022 and February 2023, respectively. *N. lappaceum* was identified by comparison with a voucher specimen (SCA 1995–403) at the Limbe Botanic Garden, with the kind assistance of M. Ndive Elias (Botanist). *H. thebaica* was identified by comparison with material deposited by Geerling C4624 of the voucher specimen of herbarium collection No. 36296SRF/Cam at the Cameroon National Herbarium (HNC, Yaounde, Cameroon), with the help of M. Eric Ngansop (Botanist).

The gathered plant parts were dried in a shelter away from direct sunlight. Then, they were ground to obtain a fine powder. The weight of each air-dried plant powder was measured (*N. Lappaceum* bark 291 g, leaves 334 g, peels 170.8 g, seeds 108.8 g, and *H. thebaica* fruits 500 g). The powders were then macerated using methanol (1:3 w/v) for 48 hours, with the macerate being mixed three times a day to increase the extraction yield. The macerate was filtered using Whatman filter paper grade 1. The same procedure was repeated twice with the plant residues. Finally, the overall filtrate was concentrated in a rotary evaporator (BUCHI Rotavapor R-200) at reduced pressure and temperature (<45˚C) to obtain a crude methanolic extract. The residual solvent was removed by drying at 35–40˚C in an oven, and the extracts were stored at 4˚C for further use. The extraction yield was determined using the following formula:

$$\text{Yield} = (\text{weight of crude extract obtained} / \text{weight of air} - \text{dried powder used}) \times 100$$

## Chemicals

Dimethyl-sulfoxide (DMSO) 0.2% (Pure Pharma Grade, USP) and TWEEN-20 (Kremer Pigmente Gmbh and CoKG) were used to dissolve plant extracts. para-Iodonitrotetrazolium chloride (INT) (Sigma-Aldrich, Germany) was used as a microbial growth indicator. Five commonly prescribed antibiotics from different classes were employed for plant extract/antibiotic combination assays, including ciprofloxacin, cefixime, imipenem, tetracycline, and vancomycin (Sigma-Aldrich, Germany).

## Antibacterial assay

**Microorganisms and culture media.** The microorganisms studied in this research are critical-class priority pathogens used to identify new antibacterial agents [3], including *Enterobacter aerogenes* (EA3 and EA27), *Staphylococcus aureus* (MRSA4and MRSA6), *Klebsiella pneumoniae* (ATCC11296 and KP63), *Pseudomonas aeruginosa* (PA01 and PA124), and *Escherichia coli* (ATCC10536 and AG100). The resistance profile of the isolates and strains used in the present study were previously reported [18, 28], and their general characteristics are presented (S1 Table). They were the American Type Culture Collection (ATCC) and clinical laboratory collections. All bacterial isolates were cultured on Mueller Hinton agar (MHA) (Liofilchem S.r.l., Italy) and the microdilution testing was done using Mueller Hinton broth (MHB) (Titan Biotech Ltd., India) to determine the test samples' minimum inhibitory concentration (MIC), minimum bactericidal concentration (MBC), and activity increase factor (AIF) (following combination assays). The MHB and MHA were prepared according to the manufacturer's instructions under strict aseptic conditions.

**Iodonitrotetrazolium chloride colorimetric test for minimum inhibitory concentration and minimal bactericidal concentration determinations.** The minimum inhibitory concentration (MIC) and minimal bactericidal concentration (MBC) are two methods used to determine the antibacterial properties of a substance. To test these, the microplate serial dilution method technique was used, and iodonitrotetrazolium chloride (INT) served as a bacterial growth indicator [28]. The widest spectrum of antibacterial activity of imipenem against both gram-positive and gram-negative bacteria [29] justifies its selection as a reference drug in the current study. To prepare the bacterial inoculum, bacterial colonies were collected from 18 to 24-hour-old bacterial culture and introduced into 10 mL of sterile distilled water. The resulting suspension solution was compared with the turbidity of a standard 0.5 McFarland solution to obtain a bacterial suspension of $1.5 \times 10^8$ CFU/mL. This bacterial suspension was diluted with MHB to yield an inoculum solution corresponding to $2 \times 10^6$ CFU/mL. The extracts of *N. lappaceum* (peels, bark, leaves, seeds) and *H. thebaica* (fruits) were dissolved in 10% DMSO/MHB (final concentration of DMSO in the solution was $\leq 2.5\%$), and then a series of two-fold dilutions were made in a 96-well microplate. After that, 100 μL of inoculum ($2 \times 10^6$ CFU/mL) prepared in MHB was added to each well. For quality control, imipenem was used as a reference drug (positive control), and wells containing the vehicle (DMSO 2.5%) were used as the negative control (no antibacterial activity was detected with DMSO $\leq 2.5\%$). The plates were then covered with a sterile sealer and incubated at 37˚C for 18–24 hours. The MIC of each sample was defined as the lowest concentration of the sample that completely inhibited bacterial growth, which was detected following the addition of 40 μL INT (0.2 mg/mL) and incubated at 37˚C for 30 min. Viable bacteria reduced the yellow dye to pink.

The MBC of the samples was determined by pipetting 50 μL of the suspensions from the wells that did not show any growth after incubation during the MIC assays to a new 96-well microplate containing 150 μL of fresh broth per well. The plate was further re-incubated at 37˚C for 48 hours, followed by the addition of 40 μL of 0.2 mg/mL INT. The MBC was defined

as the lowest concentration of samples that completely inhibited the growth of bacteria. Each assay was performed in triplicate and repeated three times to ensure accurate results. The bactericidal or bacteriostatic effect of the test extracts was determined by calculating the MBC/MIC ratio.

**Plant extract/antibiotic combination assay.** A combination assay of test herbals was performed with selected antibiotics (ciprofloxacin, cefixime, imipenem, tetracycline, and vancomycin). The assay was carried out in the same manner as the MIC test described above, with the difference that successive dilutions were performed in the presence of antibiotics [17–22]. Fifty microliters of extract at sub-inhibitory concentrations were introduced into each well, followed by adding 50 μL of bacterial inoculum at a $4 \times 10^6$ CFU/mL density. The activity of the crude extracts at sub-inhibitory concentrations (MIC/2, MIC/4, MIC/8, and MIC/16) was evaluated by a preliminary test performed on the most resistant strain (MRSA4) to select appropriate sub-inhibitory concentrations. Extracts that showed significant activity at these concentrations were retained and tested further on bacterial strains resistant to the selected antibiotics. The positive control was represented by a column of wells containing the antibiotic and inoculum without the extract. The Activity Increase Factor (AIF) or potentiation of susceptibility (fold) was calculated using the following formula:

$$AIF = MIC\ ATBalone/MIC\ ATBassociation$$

## Antibacterial modes of action

**Action of *N. lappaceum* leaf extract on the bacterial cell membrane integrity.** The experiment aimed to investigate the impact of *N. lappaceum* leaf extract on the membrane integrity of bacteria. This was done by measuring the rate of absorbance at 260 nm of cytosolic contents released into the suspension. The experiment followed the method described by Sampathkumar et al. [30], with slight modifications [31]. Young colonies (18–24 hours old) of bacteria were collected from a fresh culture on MHA medium, and a suspension of $1 \times 10^6$ CFU was prepared. The bacteria were then treated with *N. lappaceum* leaf extract at different concentrations (0.5xMIC, MIC, and 2xMIC), after which they were incubated at 37°C for 12 hours. Afterward, the samples were centrifuged, and the absorbance of the supernatant was measured at 260 nm for the control and treated cells using a spectrophotometer (Thermo Scientific, Langenselbold, Germany). A tube containing the inoculum and DMSO was used as a negative control. Each assay was performed in triplicate. An increase in absorbance at 260 nm indicated a release of intracellular contents (DNA, RNA), evidence of an alteration of the integrity of the plasma membrane.

**Action of *N. lappaceum* leaf extract on the bacterial H$^+$-ATPase-mediated proton pumping.** This mode of action was performed after acidification of the external medium of the bacteria using a pH electrode [31, 32]. Bacterial cultures were dissolved in 20 mL of MHB and incubated at 37°C while shaking for 18 hours. Bacterial aliquots of 1 mL of this pre-culture media were taken and added to the medium to obtain a final volume of 100 mL (1/100 v/v dilution). This mixture was then re-incubated at 37°C for 18 hours while shaking. One hundred milliliters of this bacterial culture was centrifuged at 3000 rpm for 10 minutes at 4°C. The recovered pellets were rinsed with sterile distilled water and then dissolved in 50 mL of 50 mM KCl. The resulting bacterial suspension was stored at 4°C for 18 hours (for glucose restriction), after which the pH was adjusted to 6.48 by adding HCl or NaOH solution. Then, 0.5 mL of the test extract was added to 4 mL of the bacterial culture ($1.5 \times 10^8$ CFU/ml), and the mixture was incubated at 37°C for 10 min. After that, 0.5 mL of 20% glucose was added to initiate the acidification of the medium. DMSO (2.5% v/v) was used as the negative control. The pH values of the tested samples were read at room temperature (25°C) every 10 min for 60 min, using a pH

meter (Thermo Scientific, Langenselbold, Germany). The pH = f (t) curve was plotted using Microsoft Excel 2016. If the medium becomes more acidic, the curves will decrease. On the other hand, if the medium becomes more basic, the curves will rise. This indicates that the extract is preventing the release of $H^+$ ions by the bacteria into the system.

## Qualitative phytochemical analysis

The bark, seeds, peels, and leaves of *N. lappaceum* and the fruits of *H. thebaica* were qualitatively screened for the presence of bioactive secondary metabolites. Standard methods were used to detect the presence of the following: alkaloids (Dragendorf test), saponins (frothing), Steroids (Lieberman- Burchard test), tannins (ferric chloride test), flavonoids (cyanide test), phenols, and cardiac glycosides [33, 34].

## Data interpretation/analysis

The results of the antibacterial activity of test herbals were interpreted based on the most recent cutoff points [35–37]. Generally, the antibacterial activity of plant extracts against Enterobacteria (*E. coli*, *E. aerogenes*, and *K. pneumoniae*) was considered outstanding if MIC $\leq$ 8 μg/mL, excellent if 8 < MIC $\leq$ 64 μg/mL, very good if 64 < MIC $\leq$ 128 μg/mL, good if 128 < MIC $\leq$ 256 μg/mL, average if 256 < MIC $\leq$ 512 μg/mL, weak if 512 < MIC $\leq$ 1024 μg/mL, and not active if MIC values > 1024 μg/mL [35]. The established cutoff point for the antibacterial activity of herbals towards *P. aeruginosa* was used as follows: outstanding activity when MIC $\leq$ 32 μg/mL; excellent activity when 32 < MIC $\leq$ 128 μg/mL; very good activity when 128 < MIC $\leq$ 256 μg/mL; good activity when 256 < MIC $\leq$ 512 μg/mL, average activity when 512 < MIC $\leq$ 1024 μg/mL, weak activity or not active when MIC values >1024 μg/mL [36]. The cut-off values for the classification of botanicals towards the Gram-positive bacteria (*S. aureus*) were as follows: outstanding activity (MIC $\leq$ 8 μg/mL), excellent activity (8 < MIC $\leq$ 40 μg/mL), very good activity (40 < MIC $\leq$ 128 μg/mL), good activity (128 < MIC $\leq$ 320 μg/mL), average activity (320 < MIC $\leq$ 625 μg/mL), weak activity (625 < MIC $\leq$ 1024 μg/mL), and not active (MIC values > 1024 μg/mL) [37]. The bactericidal or bacteriostatic effect of botanicals was determined using the MBC/MIC ratio, where MBC/MIC $\leq$4 was considered bactericidal and MBC/MIC >4 was considered bacteriostatic [38]. The resulting interactions between plant extracts and antibiotics were further interpreted using the following classification: antagonism if the activity increase factor (AIF) < 0.25, indifference if 0.25 < AIF < 2, and synergy if AIF $\geq$ 2 [39]. The data for the modes of action were presented as mean ± standard deviation of three replicates.

## Results

### Antibacterial activity

The antibacterial potential of the methanol extracts from different parts of *N. lappaceum* and fruits of *H. thebaica* was evaluated against ten isolates and strains of clinically relevant bacteria. The MIC and MBC of the extracts were determined (Table 1). The bactericidal or bacteriostatic nature of the extracts was calculated by finding the ratio of MBC to MIC. The antibacterial activity of the extracts varied depending on the bacterial strains, and the MIC results ranged from 128 to 2048 μg/mL. The bark, leaf, and peel extracts of *N. lappaceum* were found to be effective against all studied MDR bacteria, whereas the seed extract inhibited 9 out of 10 strains. The fruit extract of *H. thebaica* was less active, with MIC values obtained for only 6 out of 10 pathogens. The leaf extract of *N. lappaceum* was found to be the most active, with MICs ranging between 128 to 1024 μg/mL on all studied MDR strains and clinical isolates. The

**Table 1. Minimum inhibitory concentrations (MIC) and Minimum bactericidal concentrations (MBC) of *Nephelium lappaceum*, *Hyphaene thebaica* methanol extracts (µg/mL), and the reference drug (imipenem, in µg/mL) on the studied multidrug-resistant bacterial strains.**

| Bacteria | | MIC(MBC) of plant extracts and the reference antibiotic (in µg/mL) | | | | | |
|---|---|---|---|---|---|---|---|
| | | *N. lappaceum* | | | | *H. thebaica* | Reference drug |
| | | Bark | Leaves | Fruits peel | Seeds | Fruits | Imipenem |
| *S. aureus* | MRSA4 | 2048(-) | **512**(-) | 1024(1024) R = 1 | >2048(-) | >2048(-) | 16(256) |
| | MRSA6 | >2048(-) | **512**(-) | **512**(-) | 1024(-) | 256(-) | 16(16) |
| *P. aeruginosa* | PA01 | 512(512) R = 1 | 1024 (2048) R = 2 | **512**(2048) R = 4 | **512**(-) | >2048(-) | 16(512) |
| | PA124 | >2048(-) | **256**(512) R = 2 | 1024(-) | 2048(-) | 1024(-) | 16(512) |
| *E. coli* | AG100 | **512**(-) | **512**(-) | 2048(-) | 1024(-) | 2048(-) | 16(256) |
| | ATCC10536 | >2048(-) | **256**(2048) R = 4 | 1024(-) | **512**(512) R = 1 | >2048(-) | 16(128) |
| *K. pneumoniae* | KP63 | 1024(-) | **256**(2048) R = 4 | **512**(-) | **512**(512) R = 1 | 512(-) | 16(256) |
| | ATCC11296 | >2048(-) | 1024(-) | 1024(-) | 1024 | 2048(-) | 16(16) |
| *E. aerogenes* | EA3 | 1024(-) | 1024(-) | **512**(-) | **512**(512) R = 1 | >2048(-) | 16(32) |
| | EA27 | **128**(-) | **128**(2048) R = 8 | **512**(1024) R = 2 | 1024(-) | **512**(512) R = 1 | 16(16) |

MBC Minimum Bactericidal Concentration; MIC Minimum Inhibitory Concentration; R MBC/MIC ratio; Bold Significant activity (MIC); R≤4 Bactericidal effect; R>4 Bacteriostatic effect;—MBC value > 2048 µg/mL.

lowest MIC value of 128 µg/mL (high activity) was obtained with bark and leaf extracts of *N. lappaceum* on *E. aerogenes* EA27. Except for the leaf extract of *N. lappaceum*, all test extracts displayed MBCs on less than four microorganisms. Interestingly, all MBC/MIC ratios calculated were ≤ 4. MBC = MIC was obtained with bark and peel extracts of *N. lappaceum* on *P. aeruginosa* PA01 and *S. aureus* MRSA4, respectively. A similar effect was noted with the seed of the plant on *E. coli* and *K. pneumoniae* KP63, and the fruits of *H. thebaica* on *E. aerogenes* EA27 (Table 1).

## Combination testing

A preliminary test was conducted to evaluate the combination of plant extracts with commonly used antibiotics against the most resistant strain, MRSA4, at different concentrations. The results showed that the combination of plant extracts and antibiotics was most effective at MIC/2 and MIC/4 concentrations (Tables 2 and 3). These concentrations were then tested on extended MDR strains. The herbal extracts showed a remarkable ability to synergize with commonly used antibiotics like tetracycline, vancomycin, imipenem, ciprofloxacin, and cefixime

**Table 2. Preliminary combination assay of *H. thebaica* fruits at sub-inhibitory concentrations (MIC/2, MIC/4, MIC/8, MIC/16) with antibiotics against methicillin-resistant *Staphylococcus aureus* (MRSA4).**

| ATB | MIC 0 | *H. thebaica* fruits (AIF) | | | |
|---|---|---|---|---|---|
| | | MIC/2 | MIC/4 | MIC/8 | MIC/16 |
| CEF | 128 | 32(**4**) | 64(**2**) | 128(**1**) | 128(**1**) |
| TET | 16 | 0.5(**8**) | 0.5(**8**) | - | 4(**4**) |
| VAN | 256 | 64(**4**) | 64(**4**) | 128(**2**) | 128(**2**) |
| IMP | 16 | 8(**2**) | 8(**2**) | 8(**2**) | 8(**2**) |
| CIP | 8 | 0.5(**4**) | 2(**4**) | 4(**2**) | 4(**2**) |

- No reaction observed; CEF Cefepime; TET Tetracycline; VAN Vancomycin; IMP Imipenem; CIP Ciprofloxacin; MIC Minimal Inhibitory Concentration; ATB Antibiotics; Bold Synergy; AIF Activity Increase Factor. MIC 0 Minimal inhibitory concentration of the antibiotic alone.

**Table 3. Preliminary combination assay of *N. lappaceum* extract at sub-inhibitory concentrations (MIC/2, MIC/4, MIC/8, MIC/16) with conventional antibiotics on methicillin-resistant *Staphylococcus aureus* (MRSA4).**

| ATB | MIC 0 | *N. lappaceum* extracts (AIF) | | | | | | | | | | | | | | | |
|---|---|---|---|---|---|---|---|---|---|---|---|---|---|---|---|---|---|
| | | Bark | | | | Leaves | | | | Peels | | | | Seeds | | | |
| | | MIC/2 | MIC/4 | MIC/8 | MIC/16 | MIC/2 | MIC/4 | MIC/8 | MIC/16 | MIC/2 | MIC/4 | MIC/8 | MIC/16 | MIC/2 | MIC/4 | MIC/8 | MIC/16 |
| CEF | 128 | 64 (**2**) | 128 (1) | 128 (1) | 128 (1) | 16 (**8**) | 32 (**4**) | 128 (1) | 128 (1) | 8 (**16**) | 16 (**8**) | 32 (**4**) | 64 (**2**) | 64 (**2**) | 64 (**2**) | 128 (1) | 128 (1) |
| TET | 16 | 0.5(**8**) | 0.5 (**8**) | 1 (**16**) | 1 (**16**) | 0.25 (**4**) | 0.5(**8**) | 1 (**16**) | 1 (**16**) | 1 (**16**) | 2 (**8**) | 8(**2**) | 16(1) | 4 (**4**) | 4 (**4**) | 16 (1) | - |
| VAN | 256 | 64 (**4**) | 256 (1) | 256 (1) | 256 (1) | 32 (**8**) | 64 (**4**) | 256 (1) | 256 (1) | 8 (**32**) | 8 (**32**) | 32 (**8**) | 64 (**4**) | 128 (**2**) | 256 (1) | 256 (1) | 256 (1) |
| IMP | 16 | 2 (**8**) | 2 (**8**) | 2 (**8**) | 4 (**4**) | 2 (**8**) | 2 (**8**) | 2(**8**) | 16 (1) | 16 (1) | 16 (1) | 16 (1) | 16 (1) | 16 (1) | 16 (1) | 16 (1) | 16 (1) |
| CIP | 8 | 2 (**4**) | 2 (**4**) | 8 (1) | 8 (1) | 0.125 (1) | 0.25 (**2**) | 0.5 (**4**) | 2 (**4**) | 0.06 (0.5) | 1 (**8**) | 8 (1) | 8 (1) | 4 (**2**) | 4 (**2**) | 4 (**2**) | 4 (**2**) |

- No reaction observed; CEF Cefepime; TET Tetracycline; VAN Vancomycin; IMP Imipenem; CIP Ciprofloxacin; MIC Minimal Inhibitory Concentration; ATB Antibiotics; Bold Synergy; AIF Activity Increase Factor. MIC 0 Minimal inhibitory concentration of the antibiotic alone.

(Table 4). The leaf extract of *N. lappaceum* was found to be particularly effective, potentiating all antibiotics (100%) at MIC/2 and MIC/4 when tested on *K. pneumoniae*. Similar results were obtained with the peel extract of *N. lappaceum* on *K. pneumoniae* and *E. coli*. *H. thebaica* also improved the efficacy of all antibiotics at MIC/2 on *K. pneumoniae* and showed positive results with other strains (Table 4). No antagonistic interactions were recorded, and the seed extract of *N. lappaceum* showed no interaction with antibiotics on *P. areruginosa* PA01. The isobologram representation of synergic points of *N. lappaceum* leaf and peel extracts combined with standard antibiotics against selected highly resistant bacteria at MIC/2 and MIC/4 are presented in Figs 1 and 2.

## Alteration of the bacterial cell membrane integrity

After treating *E. aerogenes* EA27 with the leaf extract of *N. lappaceum* at 0.5×MIC, MIC, and 2×MIC, there was an increase in OD 260 nm that was dependent on the concentration. The O.D. increased up to 0.05 from 0.01, indicating no significant change after 60 minutes of exposure to the extract as compared to the control (Fig 3).

## Inhibition of H$^+$-ATPase-mediated proton pumping

The study aimed to evaluate the ability of *N. lappaceum* leaf extract to inhibit H$^+$-ATPase-mediated proton pumping in *E. aerogenes* EA27 over 60 minutes. The extract was tested at 0.5×MIC, MIC, and 2×MIC concentrations, and a significant and concentration-dependent reduction of pH was observed compared to the control (Fig 4).

## Phytochemical analysis

The herbal extracts were screened using standard methods, and the results are shown in Table 3. Secondary metabolites were found in different proportions depending on the plant extract. Saponins, phenols, and tannins were present in all extracts, while only the leaf extract of *N. lappaceum* contained steroids. Additionally, alkaloids were identified only in the seed extract of *N. lappaceum*. The extraction yield of the different herbals ranged from 0.83% for *H. thebaica* fruits to 10.61% for seeds of *N. lappaceum* (Table 5).

**Table 4. Effects of the association of *N. lappaceum* and *H. thebaica* extracts at sub-inhibitory concentrations (MIC/2 and MIC/4) with conventional antibiotics against selected multidrug-resistant bacterial strains.**

| ATB | *N. lappaceum* | | | | | | | | *H. thebaica* | | MIC 0 | BACTERIA |
|---|---|---|---|---|---|---|---|---|---|---|---|---|
| | Bark | | Leaves | | Peels | | Seeds | | Fruits | | | |
| | MIC/2 | MIC/4 | MIC/2 | MIC/4 | MIC/2 | MIC/4 | MIC/2 | MIC/4 | MIC/2 | MIC/4 | | *K. pneumoniae*ATCC11296 |
| Cefixime | 4(**32**) | 128(1) | 1(**128**) | 64(2) | 8(**16**) | 128(**1**) | 128(**1**) | 128(1) | 16(**8**) | 16(**8**) | 128 | |
| Tetracycline | 0.25(**128**) | 0.25(**128**) | 0.25(**128**) | 4(8) | 4(8) | 8(4) | 8(4) | 8(4) | 1(**32**) | 4(8) | 32 | |
| Vancomycin | 256(1) | 256(1) | 2(**128**) | 2(**128**) | 2(**128**) | 2(**128**) | 256(**1**) | 256(1) | 128(**2**) | 128(**2**) | 256 | |
| Imipenem | 1(**32**) | 8(4) | 1(**32**) | 8(4) | 4(8) | 16(2) | 32(**1**) | 32(1) | 16(2) | 32(1) | 32 | |
| Ciprofloxacin | 2(**4**) | 2(**4**) | 0.25 (**32**) | 0.25(**32**) | 4(2) | 8(**1**) | 8(**1**) | 8(1) | 4(2) | 4(2) | 8 | |
| | 80% | 60% | 100% | 100% | 100% | 60% | 20% | 20% | 100% | 80% | PPR | |
| Cefixime | 128(1) | 128(1) | 1(**128**) | 64(2) | 128(1) | 128(1) | 128(1) | 128(1) | 32(**4**) | 128(1) | 128 | *E. aerogenes*EA3 |
| Tetracycline | 1(**16**) | 4(4) | 0.25 (**4**) | 4(4) | 8(2) | 8(2) | 8(2) | 8(2) | 4(4) | 4(**4**) | 16 | |
| Vancomycin | 128(**2**) | 128(**2**) | 8(**32**) | 16(**16**) | 4(**64**) | 32(**8**) | 256(1) | 256(1) | 256(1) | 256(1) | 256 | |
| Imipenem | 8(4) | 32(1) | 1(**32**) | 4(8) | 8(4) | 16(2) | 16(2) | 32(1) | 8(4) | 16(2) | 32 | |
| Ciprofloxacin | 0.5(**4**) | 0.5(**4**) | 0.0625(0.5) | 0.0625 (0.5) | 0.0625 (0.5) | 0.125(1) | 4(2) | 4(2) | 4(2) | 4(2) | 8 | |
| | 80% | 60% | 80% | 80% | 60% | 60% | 60% | 60% | 40% | 80% | PPR | |
| Cefixime | 64(**2**) | 128(1) | 8(**16**) | 128(1) | 128(1) | 128(1) | 128(1) | 128(1) | 32(**4**) | 64(**2**) | 128 | *P. aeruginosa*PA01 |
| Tetracycline | 0.25 (**4**) | 0.5 (**8**) | 4(4) | 8(2) | 8(2) | 32(0.5) | 16(1) | 16(1) | 8(2) | 8(2) | 16 | |
| Vancomycin | 256(1) | 256(1) | 128(**2**) | 128(**2**) | 128(**2**) | 128(**2**) | 256(1) | 256(1) | 256(1) | 256(1) | 256 | |
| Imipenem | 32(1) | 32(1) | 8(4) | 16(2) | 16(2) | 16(2) | 32(1) | 32(1) | 4(**8**) | 4(**8**) | 32 | |
| Ciprofloxacin | 8(1) | 8(1) | 2(**4**) | 8(1) | 8(1) | 8(1) | 8(1) | 8(1) | 8(1) | 8(1) | 8 | |
| | 40% | 20% | 100% | 60% | 60% | 40% | 0% | 0% | 60% | 60% | PPR | |
| Cefixime | 128(1) | 128(1) | 64(2) | 128(1) | 4(**32**) | 64(2) | 128(1) | 128(1) | 16(**8**) | 64(**2**) | 128 | *E. coli*AG100 |
| Tetracycline | 4(4) | 8(2) | 4(4) | 4(4) | 2(8) | 8(2) | 16(2) | 16(2) | 4(4) | 4(4) | 16 | |
| Vancomycin | 256(1) | 256(1) | 64(4) | 64(4) | 2(**128**) | 64(4) | 256(1) | 256(1) | 256(1) | 256(1) | 256 | |
| Imipenem | 16(2) | 16(2) | 8(4) | 16(2) | 0.25(**8**) | 16(2) | 16(2) | 16(2) | 16(2) | 16(2) | 32 | |
| Ciprofloxacin | 2(**4**) | 2(**4**) | 0.0625 (0.5) | 0.5(**4**) | 0.0625 (0.5) | 0.25 (**2**) | 2(**4**) | 4(2) | 2(**4**) | 4(2) | 8 | |
| | 60% | 60% | 80% | 80% | 80% | 100% | 60% | 60% | 80% | 80% | PPR | |

Values in bracket Activity Increase Factor. MIC Minimum Inhibitory Concentration; MBC Minimum Bactericidal Concentration; ATB Antibiotics; Bold Synergy. PPR Percentage of potentiation recorded. MIC 0 Minimal inhibitory concentration of the antibiotic alone.

## Discussion

Infections caused by MDR bacteria pose a serious threat to public health. Recent studies have suggested that phytochemicals, naturally occurring compounds found in plants, could help manage these difficult-to-treat infections [6]. To explore this possibility, we researched the antibacterial potential of two edible plants, *N. lappaceum* and *H. thebaica*, against MDR bacterial strains and clinical isolates. We also investigated the possible modes of action of the most effective extract and evaluated the ability of the test herbals to enhance the efficacy of commonly prescribed antibiotics, including cefixime, ciprofloxacin, tetracycline, imipenem, and vancomycin. The microorganisms we tested included gram-negative bacteria (*E. coli*, *E. aerogenes*, *P. aeruginosa*, and *K. pneumoniae*) and a gram-positive bacterium (methicillin-resistant *S. aureus*). All of the strains and clinical isolates used exhibited drug-resistant phenotypes (S1 Table) and are among the WHO's critical-class priority pathogens for the discovery of new and effective antibacterial agents [3]. This study, therefore, has the potential to provide a foundation for the development of potent antibacterial phytochemicals to combat difficult-to-treat infections.

The antibacterial activity of test extracts was ascertained through the determination of MICs, which varied from 128 to 2048 μg/mL. The best activity was obtained with the leaf

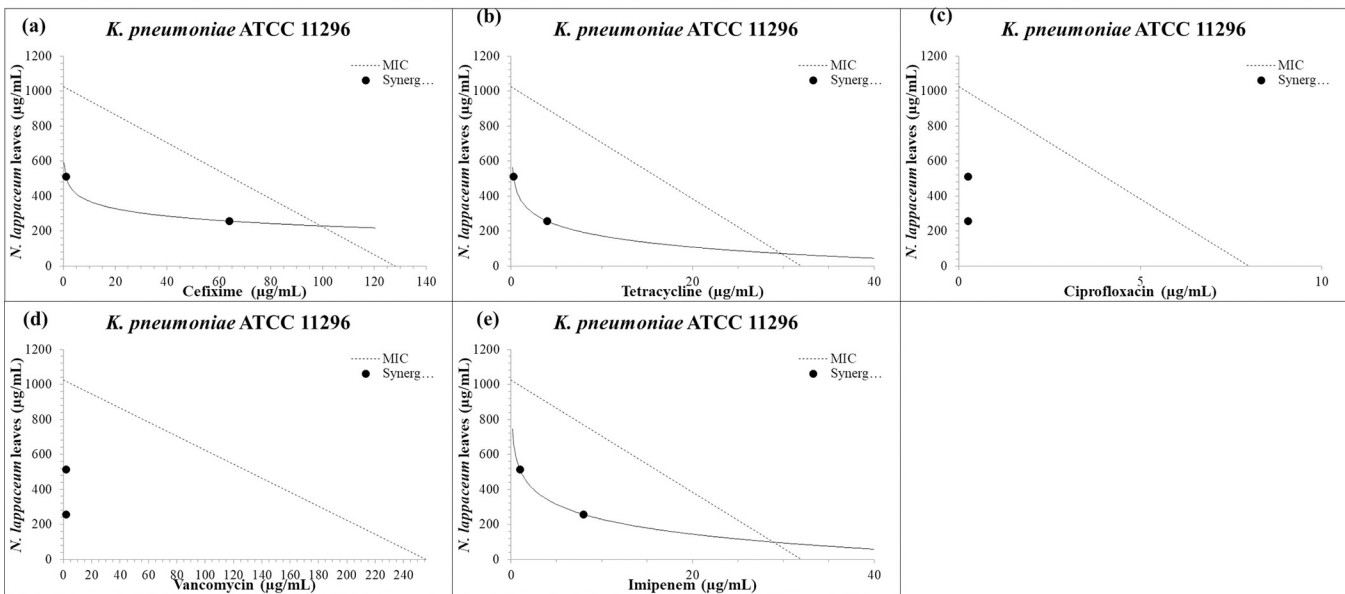

**Fig 1. Isobologram representation of synergic points of *N. lappaceum* leaf extract combined with standard antibiotics against *K. pneumoniae* ATCC11296 at MIC/2 and MIC/4.** Synergistic interactions were more pronounced in combination with ciprofloxacin (c) and vancomycin (d), then followed by tetracycline (b), imipenem (e), and ultimately cefixime (a).

methanol extract of *N. lappaceum*, which depicted a MIC range between 128 and 1024 μg/mL in all studied MDR bacterial strains and clinical isolates. Furthermore, the bark and peel of *N. lappaceum* showed inhibitory effects on all tested bacteria, whereas the seeds displayed activity on 9 out of 10 strains. The lowest MIC (128 μg/mL) was recorded with the leaves and bark of *N. lappaceum* on *E. aerogenes* EA27 (Table 1). The fruit extract of *H. thebaica* produced the

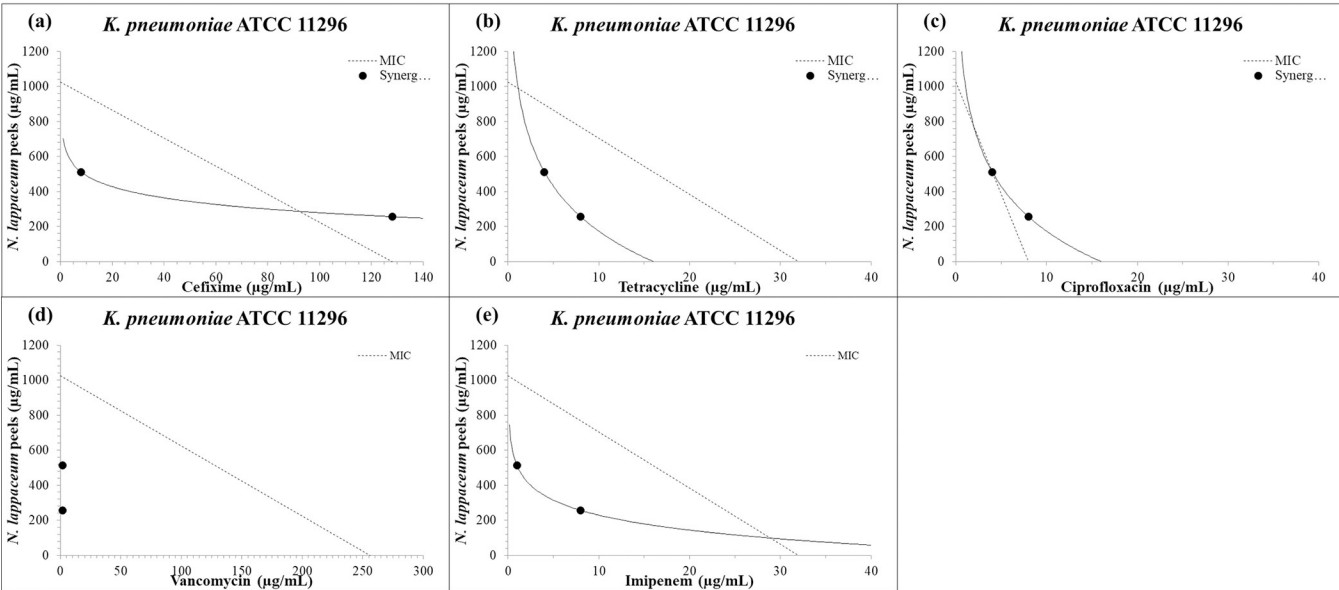

**Fig 2. Isobologram representation of synergic points of *N. lappaceum* peel extract combined with standard antibiotics against *K. pneumoniae* ATCC11296 at MIC/2 and MIC/4.** Synergistic interactions were more pronounced in combination with vancomycin (d), imipenem (e), and tetracycline (b), then followed by cefixime (a), and ultimately ciprofloxacin (c).

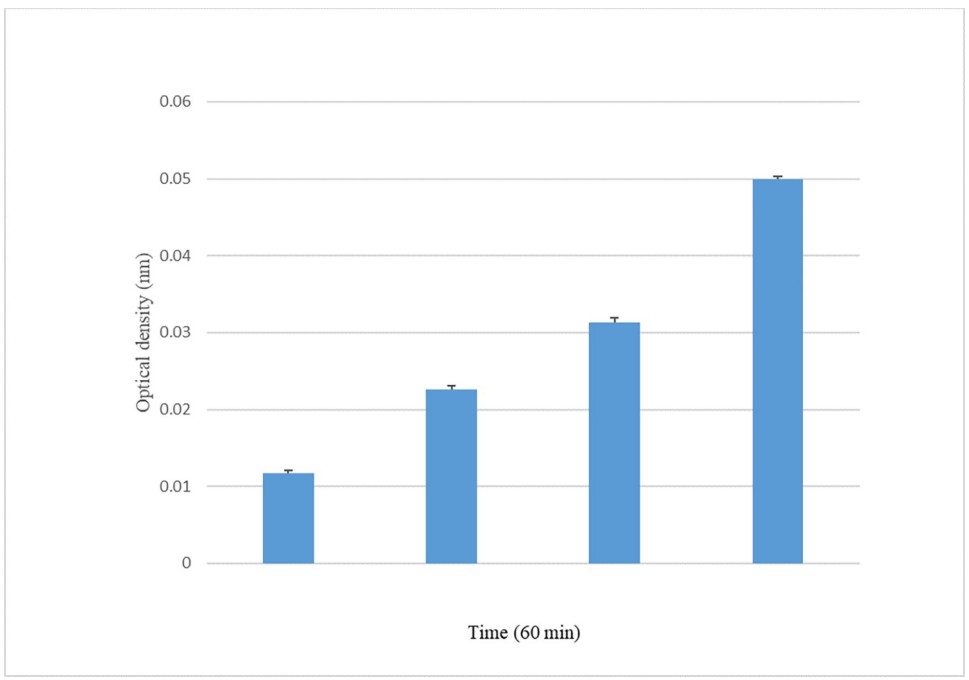

**Fig 3. Effect of the leaf methanol extract of *N. lappaceum* on cell membrane integrity of *E. aerogenes* EA27.**

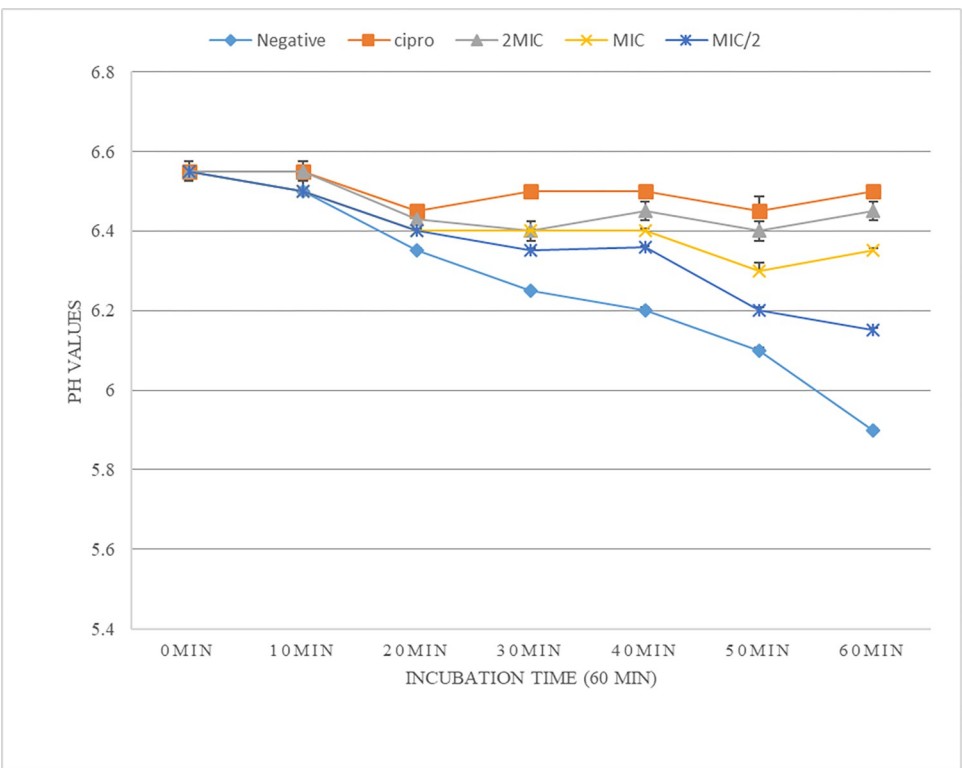

**Fig 4. Effect of the leaf methanol extract of *N. lappaceum* on H$^+$-ATPase-mediated proton of *E. aerogenes* EA27.**

**Table 5. Phytochemical composition of the studied plant extracts with their corresponding percentage yields.**

| Plant secondary metabolites | *N. lappaceum* | | | | *H. thebaica* |
|---|---|---|---|---|---|
| | Bark | Leaves | Peels | Seeds | Fruits |
| Steroids | - | +++ | - | - | - |
| Alkaloids | - | - | - | ++ | - |
| Cardiac glycoside | - | + | +++ | - | ++ |
| Phenolic | +++ | ++ | +++ | + | +++ |
| Tannins | ++ | +++ | +++ | + | ++ |
| Flavonoids | ++ | - | - | + | ++ |
| Saponins | +++ | ++ | ++ | + | ++ |
| % extraction yield | 2.19 | 1.72 | 7.75 | 10.61 | 0.83 |

+++ (60–75) %; ++ (50–55%); + (30–45%);—(0–10%) or absent.

least spectrum of activity, acting on 6 out of 10 bacteria. Interestingly, MIC of 256 μg/mL was recorded with *H. thebaica* on MRSA6 and 512 μg/mL, respectively, on *K. pneumoniae* KP63 and *E. aerogenes* EA27 (Table 1). Following established threshold values for the classification of antibacterial activity of botanicals [35–37], the antibacterial properties of *N. lappaceum* and *H. thebaica* revealed in this study range from very good to weak. The findings are important considering the bacteria's MDR nature. These results indicate the potential of these extracts to combat difficult-to-treat bacterial infections, making them an important area of further research. The differences in antibacterial effects of different parts of the test plants can be explained by the variations in the content of phytochemicals, both qualitatively and quantitatively, and the potential interactions among the phytoconstituents. Additionally, the variations in the MIC and MBC of the extracts on different bacterial strains may be due to the inherent characteristics and specific resistance features of each pathogen. Furthermore, the multiple phyto-compounds in each extract may have different antibacterial targets. Antibacterial agents are commonly categorized as bactericidal or bacteriostatic. It's noteworthy that all MBC/MIC ratios calculated were $\leq 4$, indicating the bactericidal effects of the test extracts [38]. Furthermore, MBC = MIC was observed with bark and peel extracts of *N. lappaceum* on *P. aeruginosa* PA01 (MIC = MBC = 512 μg/mL) and *S. aureus* MRSA4 (MIC = MBC = 1024 μg/mL), respectively. A similar effect was noted with the seed of *N. lappaceum* on *E. coli* ATCC10536 and *K. pneumoniae* KP63, and the fruits of *H. thebaica* on *E. aerogenes* EA27, all displaying MIC = MBC = 512 μg/mL (Table 1). This indicates the ability of the specified extracts to produce drug concentrations that can eliminate 99.9% of the exposed organisms at the recorded MICs. If the MBC/MIC ratio is high (MBC/MIC > 4), it may be impossible to deliver quantities of the antibiotic that can eliminate 99.9% of the bacteria safely, and the drug is classified as bacteriostatic [38]. *N. lappaceum* is a plant that possesses remarkable antibacterial properties. The leaves of this plant were found effective against all tested MDR strains and isolates in the present study. Previous studies have also shown that *N. lappaceum* leaves are potent against MRSA and *P. aeruginosa* [25, 40], two of the most commonly encountered bacteria in healthcare settings. Other parts of the plant, such as the peels and seeds, also contain antibacterial compounds that can combat bacteria like *P. aeruginosa*, *S. aureus*, *E. coli*, and more [24, 41]. The study conducted by Asghar et al. [41] confirmed that the epicarp of *N. lappaceum* is effective against MRSA, *P. aeruginosa*, and *K. pneumoniae*. With all of this evidence, it is clear that *N. lappaceum* is a powerful natural material in the fight against bacterial infections. Discoveries in recent studies have also revealed the impressive antibacterial properties of *H. thebaica*, a plant worth exploring [42]. The fruit pulp's methanol extract displays potent antibacterial

properties against *S. aureus* and *P. aeruginosa*, with moderate activity against *E. coli*. Further tests show that the n-hexane and aqueous extracts of *H. thebaica*'s fruits demonstrated significant antibacterial activity against *K. pneumoniae* and *P. aeruginosa* [43]. These findings corroborate the activities obtained in the present study and confirm that *H. thebaica* has great potential as a natural source of new antimicrobial agents. *N. lappaceum* and *H. thebaica* could be used alone or in combination with commonly used antibiotics for superior efficacy. By considering these test herbals, we could revolutionize our approach to fighting bacteria and enhance our ability to tackle infections.

Antibiotic resistance is a serious problem that requires immediate attention. Fortunately, botanicals have shown promising results in mitigating the spread of resistance in bacteria. Recent studies have demonstrated that some botanicals possess antibiotic-modulation effects against bacteria that exhibit MDR [9–21]. Our study tested methanol extracts of *N. lappaceum* and *H. thebaica* at sub-inhibitory concentrations in combination with five commonly prescribed antibiotics. The results were remarkable (Tables 2–4). The test extracts significantly improved the activity of selected antibiotics. The most potent extract was the leaves of *N. lappaceum*, which resulted in 100% potentiation of the antibiotics against *K. pneumoniae*. The synergistic interaction can be explained by the extracts' ability to disrupt the cytoplasmic membrane, thereby increasing antibiotics' permeability and consequently the intracellular concentration. Moreover, the bacteria used in the study, particularly the Gram-negative, are known to over-express active efflux [16, 33], which is one of the main resistance strategies resulting in MDR in bacteria [4–6]. The noteworthy synergistic interactions recorded in the present work may suggest that the extracts can block the bacterial efflux machinery, acting as an efflux pump inhibitor, thereby favoring an increased concentration of antibiotics within the bacterial cell [4, 6], sufficient to kill the pathogen. Furthermore, possible interactions with key bacterial enzymes (such as beta-lactamases produced by most Gram-negative bacteria used) can inhibit their activities thereby preventing further destruction of antibiotics having beta-lactam moiety [44], such as imipenem used in the present work. These findings are significant and should be used to revitalize outdated antibiotics that are gradually losing their potency due to resistance. By combining plant extracts with commonly used antibiotics, we can enhance the effectiveness of these drugs and combat antibiotic resistance. In addition, the findings showed that there were no antagonistic interactions between the plant extracts and antibiotics. This suggests that the extracts either maintained or even improved the effectiveness of the antibiotics. These results are not only promising but also highlight the potential for enhancing antibiotic properties through synergistic interactions with plant extracts.

The diverse range of phytochemicals found in herbals allows them to act on multiple targets. They can bind to the cell membrane, cell wall, or intracellular targets such as proteins, DNA/RNA, and proton pumps. Phytochemicals need to locate specific binding receptors and action sites. Those that operate within cells must penetrate the cell membrane to identify targets within the cell [30, 31]. The present research shows that *N. lappaceum* leaf extract can disrupt the bacteria cell membrane and induce leakage of the intracellular components (DNA, RNA). This is shown by an increase in OD at 260 nm (Fig 3). The study by Phuong et al. [45] showed similar results, depicting the ability of *N. lappaceum* to disrupt the bacterial cell membrane. The extract also inhibited the $H^+$-ATPase-mediated proton pumps of *E. aerogenes* EA27, leading to a significant reduction in pH when compared to the control (Fig 4). This is an important finding as the proton pumps are responsible for maintaining a homeostatic media required for bacteria to function correctly. Inhibition of $H^+$-ATPase-dependent proton pumps can lead to a decrease in the survival of bacteria, as they require a certain level of energy in the medium for metabolism, growth, and multiplication [46]. Based on these findings, it is suggested that *N. lappaceum* leaf extract could be an effective bacterial proton pump inhibitor.

The antimicrobial properties of test extracts are generally attributed to the various bioactive secondary metabolites they contain. The presence of tannins, phenols, saponins, cardiac glycosides, and steroids was revealed in *N. lappaceum* and *H. thebaica* (Table 5), which corroborates the previous findings on the studied plants. Indeed, numerous studies have unveiled a treasure trove of phytochemicals in the *N. lappaceum* methanol extract [23, 26, 27], including potent compounds such as ellagic acid, corilagin, rutin, geraniin, galloylshikimic acid, and quercetin hexoside [45, 47]. Moreover, a plethora of other compounds, such as apigenin, gallic acid, vanillic acid, bre-vifolin carboxylic acid, pedunculagin, and theaflavin 3,3′-O-digallate, have been discovered in the extract [48, 49], making it a rich source of bioactive constituents. Furthermore, the fruits of *H. thebaica* have been found to contain an impressive array of flavonoids, steroids, terpenes, tannins, and cardiac glycosides [50], all of which contribute to their remarkable antimicrobial properties. The analysis of *H. thebaica* extract using advanced techniques has shed light on its phytochemical composition, with flavonoids, phenolic acids, and saponins emerging as the standout components [51]. Both plant species boast a lineup of potent compounds, including corilagin, rutin, ellagic acid, and quercetin, known for their exceptional antibacterial properties against MDR bacteria [6]. These findings underscore the remarkable potential of these natural sources in the quest for effective antibacterial agents. Furthermore, many plant-based antimicrobials that have well-established mechanisms of action are derived from the key phytochemical groups reported [6, 52]. Tannins, for instance, play a crucial role in inhibiting bacterial growth by affecting the cell walls, causing permeability and wrinkling. This allows tannins to penetrate the bacterial membrane, interfere with the cell's metabolism, and ultimately kill it. On the other hand, the presence of saponins leads to the leakage of enzymes and proteins from inside the cells. This bioactive component has detergent-like properties that can increase the permeability of the bacterial cell membranes, facilitating the influx of chemicals into the cell and ultimately causing damage [6, 30, 52]. The initial weight of the powder varies from one extract to another, which can account for the differences in extraction yield obtained. The variances in phytochemical profiles can be attributed to the differences in plant parts used and the inherent phytochemical composition of each plant or plant part. Certain plant parts, such as seeds and peels, may contain concentrated amounts of phytochemical agents or oils (oily extracts usually have higher weight). Furthermore, the extraction yield could be associated with the quantity of bioactive ingredients or secondary metabolites extracted from the plant. However, this alone is not adequate to produce antibacterial efficacy, since the type of phytochemical in a plant extract and the potential interactions among the phytoconstituents are also crucial factors that could influence the antibacterial efficacy of the plant extract. For example, in our study, the extract (leaves of *N. lappaceum*) with the lowest extraction yield exhibited the best antibacterial efficacy.

The current study establishes a solid experimental foundation for considering the potential of *N. lappaceum* and *H. thebaica* extracts for further investigation. This could involve exploring the cellular and molecular mechanisms of the crude extract, as well as identifying and isolating phytochemicals that target essential components of bacterial cells, such as membrane and cytoplasmic proteins, cell wall constituents (peptidoglycan), DNA, extracellular and intracellular bacterial enzymes, ribosomal function, and folate synthesis. Additionally, it would be beneficial to assess the pharmacokinetic and toxicological profile as well as the drug-likeness of key phytochemicals to identify the most promising antibacterial lead compounds from these plants for drug development.

## Conclusion

Discovering new and effective ways to fight bacterial infections is crucial nowadays. The present study has found that the extracts of *N. lappaceum* and *H. thebaica* plants have both

antibacterial and antibiotic-resistance modulatory potential. The leaf extract of *N. lappaceum* has shown significant inhibition of bacterial H$^+$-ATPase-mediated proton pumping and changes in the cell membrane integrity, suggesting possible modes of action. Furthermore, the study identified diverse classes of secondary metabolites from the plants, which might explain their antibacterial properties. These natural sources have the potential to act as adjuvants to enhance the effectiveness of antibiotics. The findings of this study are promising and suggest that these natural sources could be an important baseline for a new and effective treatment for hard-to-treat bacterial infections.

## Supporting information

**S1 Table. Common characteristics of critical-class priority bacteria used.**
(DOCX)

## Author Contributions

**Conceptualization:** Armel Jackson Seukep, Lucy M. Ayamba Ndip.

**Data curation:** Armel Jackson Seukep.

**Formal analysis:** Armel Jackson Seukep, Helene Gueaba Mbuntcha.

**Investigation:** Fula Mabel Tamambang, Valaire Yemene Matieta.

**Methodology:** Fula Mabel Tamambang, Valaire Yemene Matieta.

**Resources:** Francis Desire Tatsinkou Bomba, Victor Kuete.

**Software:** Armel Jackson Seukep.

**Supervision:** Armel Jackson Seukep, Lucy M. Ayamba Ndip.

**Validation:** Armel Jackson Seukep, Lucy M. Ayamba Ndip.

**Writing – original draft:** Armel Jackson Seukep.

**Writing – review & editing:** Armel Jackson Seukep, Helene Gueaba Mbuntcha, Victor Kuete, Lucy M. Ayamba Ndip.

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
