## [Decision Letter · Decision Letter 0]

27 Aug 2024

PONE-D-24-16310Potential of methanol extracts of Nephelium lappaceum (Sapindaceae) and Hyphaene thebaica (Arecaceae) as adjuvants to enhance the efficacy of antibiotics against critical class priority bacteriaPLOS ONE

Dear Dr. Seukep,

Thank you for submitting your manuscript to PLOS ONE. After careful consideration, we feel that it has merit but does not fully meet PLOS ONE’s publication criteria as it currently stands. Therefore, we invite you to submit a revised version of the manuscript that addresses the points raised during the review process.

**ACADEMIC EDITOR: Revision Requested**

We look forward to receiving your revised manuscript.

Kind regards,

Divakar Sharma

Academic Editor

PLOS ONE

Additional Editor Comments:

Major Revision Requested

Comments from the Editorial Team: One or more of the reviewers has recommended that you cite specific previously published works. Members of the editorial team have determined that the works referenced are not directly related to the submitted manuscript. As such, please note that it is not necessary or expected to cite the works requested by the reviewer.

Reviewers' comments:

Reviewer's Responses to Questions

**Comments to the Author**

1. Is the manuscript technically sound, and do the data support the conclusions?

Reviewer #1: Yes

Reviewer #2: Yes

Reviewer #3: Yes

Reviewer #4: Partly

Reviewer #5: No

2. Has the statistical analysis been performed appropriately and rigorously? 

Reviewer #1: I Don't Know

Reviewer #2: Yes

Reviewer #3: Yes

Reviewer #4: No

Reviewer #5: Yes

3. Have the authors made all data underlying the findings in their manuscript fully available?

Reviewer #1: No

Reviewer #2: Yes

Reviewer #3: Yes

Reviewer #4: Yes

Reviewer #5: Yes

4. Is the manuscript presented in an intelligible fashion and written in standard English?

Reviewer #1: Yes

Reviewer #2: Yes

Reviewer #3: Yes

Reviewer #4: Yes

Reviewer #5: No

5. Review Comments to the Author

Reviewer #1: Methodology:

"All bacterial isolates were cultured on Mueller Hinton agar (MHA) (Liofilchem S.r.l., Italy) and

the microdilution testing was done using Mueller Hinton broth (MHB) (Titan Biotech Ltd.,

India) to determine the test samples' minimum inhibitory concentration (MIC), minimum

bactericidal concentration (MBC), and activity increase factor (AIF) (following combination

assays)."

Q: The combination assay should be elaborated in the methodology section.

Results:

Table 1- Why MBC values are not stated for all the tested microorganism?Hence it has resulted in not have the MBC/MIC ratio values for respective bacterial isolates.

Why only one Antibiotic (Imipenem) has been used as reference antibiotic for all the tested microorganism? Explain this.

Combination Testing result should be provided as part of the results and not as Supplemental information 2 and 3.

Reviewer #2: The authors have examined the efficacy of methanolic extracts from two food plants (Nephelium lappaceum and Hyphaene thebaica), alone and in combination with antibiotics, against normal and resistant bacteria strains. Below are some comments

Result section :

• The authors do not indicate the MIC unit of the reference antibiotic (Imipenem). Is it also in (μg/mL)?

• It would be possible to use the most recent MIC classification scales, which reflect the current level of bacterial virulence in clinical practice.

• The figures shown are not numbered.

• The first figure is not numbered on the x-axis.

Reviewer #3: This work reports the Potential of methanol extracts of Nephelium lappaceum (Sapindaceae) and Hyphaene thebaica (Arecaceae) as adjuvants to enhance the efficacy of antibiotics against critical class priority bacteria

The work is meaningful and interesting.

In the "Introduction" or "Experimental Section", the authors need to provide or add a more detailed explanation about exploring the antibacterial activity of plants which in this study was used to methanol extracts. Here are some papers that can be used by the authors to find out more about antibacterial activity of plants and need to be added and cited by the authors in this manuscript:

doi: 10.1016/j.sjbs.2021.10.057

doi: 10.1155/2021/6663399

doi: 10.1038/s41598-023-32900-1

Reviewer #4: The current Manuscript Number: PONE-D-24-16310; entitled:Manuscript Title: Potential of methanol extracts of Nephelium lappaceum (Sapindaceae) and Hyphaene thebaica (Arecaceae) as adjuvants to enhance the efficacy of antibiotics against critical class priority bacteria

The current manuscript contained a good idea and i have the following comments:

- Please add statistical analysis in methods section and explain the significance along the whole manuscript.

- It essential to identify the bioactive compounds of extracts using one of the chromatographic analysis e.g : Gad chromatography and link the antimicrobial activity to these compounds.

- I highly recommend to examine antioxidant actions of extracts and compare all of them.

- Please rewrite the manuscript sections after the adding the suggesting experiments and statistics.

Reviewer #5: 1. Why do various portions of Nephelium lappaceum and Hyphaene thebaica have distinct antibacterial effects against multidrug-resistant bacteria?

2. What caused Nephelium lappaceum leaf extract to have the greatest antibacterial activity against all multidrug-resistant strains? What are its mechanisms?

3. What causes the extracts' MIC and MBC values to vary for various bacterial strains?

4. Given the synergy between plant extracts and antibiotics, how can Nephelium lappaceum and Hyphaene thebaica extracts boost antibiotic efficacy?

5. What does the MBC/MIC ratio of < 4 indicate for most extracts, and how does it affect their bactericidal vs bacteriostatic properties?

6. How does Nephelium lappaceum leaf extract's capacity to lower pH by inhibiting H⁺-ATPase relate to its antibacterial effectiveness?

7. How can certain secondary metabolites (saponins, phenols, tannins, steroids, alkaloids) affect the antibacterial activity of plant extracts, and how could their presence or absence affect results?

8. What does the absence of antagonistic interactions between plant extracts and antibiotics mean for future combination treatment methods for multidrug-resistant bacterial infections?

9. How do Nephelium lappaceum and Hyphaene thebaica phytochemical profiles and extraction yields differ, and how could they affect their antibacterial applications?

10. What further study is required to understand the cellular and molecular mechanisms of Nephelium lappaceum and Hyphaene thebaica extracts' antibacterial properties, especially their impact on bacterial cell membranes and metabolic processes?

11. The author needs to provide GC-MS data of the active molecules present in the extracts.

12. It is essential to compare the biological activity of the extract with that of the pure compound identified through GC-MS analysis. This comparison will add novelty to the study; otherwise, using extracts alone lacks significance, as there are already numerous studies (more than 1000 publications) focused on extract-based biological activities.

6. PLOS authors have the option to publish the peer review history of their article (what does this mean?). If published, this will include your full peer review and any attached files.

Reviewer #1: No

Reviewer #2: No

Reviewer #3: No

Reviewer #4: **Yes: **Mohammed Yosri

Reviewer #5: No

---

## [Author Response · Author response to Decision Letter 0]

1 Oct 2024

REPLY TO JOURNAL REQUIREMENTS:

Response: Dear Editor, the manuscript was revised to meet PLOS ONE's style requirements as displayed in the PLOS ONE style templates.

Response: Dear Editor, no permits were required for the work. Investigations were performed within the Laboratory of Biochemistry of the Faculty of Health Sciences of the University of Buea (Cameroon) led by myself, and the Laboratory of Cancer Research of the University of Dschang (Cameroon) under the chair of Prof. Kuete Victor, who is our research group leader and co-author of this submission. 

However, since the studied plants were identified on field sites, out of our leadership, the names of experts (botanists) who identified/authenticated the plant material were provided in the manuscript's ‘Plant material and extraction’ section. The plant samples are commonly edible plants, not listed as endangered species, and were collected in fresh form in the markets, requiring no special permit. 

Response: Dear Editor, we confirm that our submission contains all raw data required to replicate the results of the study. Moreover, there are no ethical or legal restrictions on sharing data. The Data Availability Statement has been added to the manuscript, before References, as follows:

‘We confirm that our submission contains all raw data required to replicate the results of the study. Moreover, there are no ethical or legal restrictions on sharing data.’ 

Response: Dear Editor, the captions for the Supporting Information files have been added at the end of the manuscript. In-text citations and the reference list were updated based on the additional information in the manuscript.

REPLY TO THE REVIEWERS’ COMMENTS:

Reviewer #1: 

Methodology:

"All bacterial isolates were cultured on Mueller Hinton agar (MHA) (Liofilchem S.r.l., Italy) and the microdilution testing was done using Mueller Hinton broth (MHB) (Titan Biotech Ltd., India) to determine the test samples' minimum inhibitory concentration (MIC), minimum bactericidal concentration (MBC), and activity increase factor (AIF) (following combination assays)."

Q: The combination assay should be elaborated in the methodology section.

Response: Dear Reviewer, thanks for the observation. The full description of the protocol for combination assay is provided in the ‘Plant extract/antibiotic combination’ section of the methodology. 

Results:

Table 1- Why MBC values are not stated for all the tested microorganisms? Hence it has resulted in not having the MBC/MIC ratio values for respective bacterial isolates.

Response: Dear Reviewer, thanks for the pertinent remark. Plant extracts were tested at 2048 µg/mL as the highest concentration. Therefore, MBC values were not stated for all extracts having a MBC > 2048 µg/mL. Hence, since exact MBC values were not recorded, this prevented us from calculating the MBC/MIC ratio. For clarification, the symbol (-) was added to Table 1 (besides MIC values) and the footnotes to indicate MBC values > 2048 µg/mL. 

Why only one Antibiotic (Imipenem) has been used as a reference antibiotic for all the tested microorganisms? Explain this.

Response: Dear Reviewer, thanks for the question. Imipenem has been chosen as the reference antibiotic for all the tested microorganisms because of its wide spectrum of antibacterial activity against both gram-negative and gram-positive aerobic bacteria, including many multi-drug resistant strains, as used in our study. 

This justification and corresponding citation were added in the MIC determination section of the methodology. 

Combination Testing results should be provided as part of the results and not as Supplemental information 2 and 3.

Response: Dear Reviewer, thanks for the constructive remark. The corresponding results from supplemental information 2 and 3 were moved to the main text of the results section. They have been renamed as Table 2 and Table 3. 

Reviewer #2: 

The authors have examined the efficacy of methanolic extracts from two food plants (Nephelium lappaceum and Hyphaene thebaica), alone and in combination with antibiotics, against normal and resistant bacteria strains. Below are some comments

Result section :

• The authors do not indicate the MIC unit of the reference antibiotic (Imipenem). Is it also in (μg/mL)?

Response: Dear Reviewer, thanks for the remark. The MIC of the reference drug Imipenem is in μg/mL, and this detail has been added to Table 1. 

• It would be possible to use the most recent MIC classification scales, which reflect the current level of bacterial virulence in clinical practice.

Response: Dear Reviewer, thanks for the constructive comments. We updated the MIC classification of the antibacterial activity of botanicals to the most recent ones. Detailed information was provided in the new section ‘Data interpretation/analysis’ of the methodology, then, consistently used in the discussion section.

• The figures shown are not numbered.

Response: Dear Reviewer, thanks for the observation. The figures have been double-checked and numbered consistently. 

• The first figure is not numbered on the x-axis.

Response: Dear Reviewer, thanks for the remark. The x-axis is numbered, however, there is a difference in the x-axis interval of the various figures to facilitate the appearance and reading of resulting interactions.

Reviewer #3: 

This work reports the Potential of methanol extracts of Nephelium lappaceum (Sapindaceae) and Hyphaene thebaica (Arecaceae) as adjuvants to enhance the efficacy of antibiotics against critical class priority bacteria.

The work is meaningful and interesting.

Response: Dear Reviewer, we are grateful for your appreciation of our work. Thanks.

In the "Introduction" or "Experimental Section", the authors need to provide or add a more detailed explanation about exploring the antibacterial activity of plants which in this study was used to methanol extracts. Here are some papers that can be used by the authors to find out more about the antibacterial activity of plants and need to be added and cited by the authors in this manuscript:

doi: 10.1016/j.sjbs.2021.10.057

doi: 10.1155/2021/6663399

doi: 10.1038/s41598-023-32900-1

Response: Dear Reviewer, thanks for the suggested references. More details were added to justify the antibacterial activity of plants in the introduction, with additional references amongst which the three references suggested. 

Reviewer #4: 

The current Manuscript Number: PONE-D-24-16310; entitled: Manuscript Title: Potential of methanol extracts of Nephelium lappaceum (Sapindaceae) and Hyphaene thebaica (Arecaceae) as adjuvants to enhance the efficacy of antibiotics against critical class priority bacteria

The current manuscript contains a good idea and I have the following comments:

- Please add statistical analysis in the methods section and explain the significance along the whole manuscript.

Response: Dear Reviewer, thanks for valuing our research and the pertinent remark. The ‘data interpretation/analysis’ section was added to the methodology section, then the significance was consistently used to analyze and discuss the results obtained. 

 - It is essential to identify the bioactive compounds of extracts using one of the chromatographic analyses e.g.: Gas chromatography and link the antimicrobial activity to these compounds.

Response: Dear Reviewer, thanks for the insightful comment. We agree with the suggestion, but due to limited equipment, we were unable to perform further phytochemical analyses such as GC-MS. However, we carried out a preliminary screening of the test samples which provided an overview of the major classes of bioactive secondary metabolites from the plants that may account for the recorded activities. The in-depth analysis of the bioactive extracts is in perspective and we expect to develop future collaboration to achieve it. 

However, in addition to the preliminary phytochemical screening we performed, we reviewed the major phytochemicals previously identified from the studied plants and highlighted the link to their antibacterial effects. We expect that this can be sufficient at this level. 

- I highly recommend to examine antioxidant actions of extracts and compare all of them.

Response: Dear Reviewer, thanks for the pertinent recommendation, we appreciate it. Although the antioxidant actions would be an important addition to the current work, we currently focus in our group on the antibacterial actions of herbals against multi-drug resistant phenotypes. However, we appreciate the suggestion and may consider it as a perspective for further biopharmaceutical examination of the plants of interest. 

- Please rewrite the manuscript sections after adding the suggested experiments and statistics.

Response: Dear Reviewer, thanks for the comments. The manuscript was revised and details about data interpretation/analysis (statistics) were added. 

Reviewer #5: 

Dear Reviewer, we especially thank you for the thorough and constructive questions. These permit us to significantly improve the discussion section of the manuscript, where most of the answers (when not yet previously explained) provided here have been added. 

1. Why do various portions of Nephelium lappaceum and Hyphaene thebaica have distinct antibacterial effects against multidrug-resistant bacteria?

Response: Dear Reviewer, thanks for the pertinent question. The differences recorded in the antibacterial effects of various parts of the test plants can be explained by the differences in phytochemicals content, both qualitatively and quantitatively, the possible interactions amongst the phytoconstituents, and the antibacterial modes of action. This explanation has been added to the discussion section.

2. What caused Nephelium lappaceum leaf extract to have the greatest antibacterial activity against all multidrug-resistant strains? What are its mechanisms?

Response: Dear Reviewer, thanks for the insightful question. Nephelium lappaceum leaf extract may contain key constituents capable of neutralizing drug-resistant components in studied bacteria. In the present study, the phytochemical analysis of the extract from Nephelium lappaceum leaves revealed the presence of steroids, phenolics, tannins, and saponin, which are classes of secondary metabolites from which well-established plant antibacterial drugs have been reported against multi-drug resistant bacteria strains.

In addition, possible mechanisms of its antibacterial actions were unveiled in our research. Indeed, the findings revealed the potential of the extract to alter the bacterial cell membrane and induce leakage of genetic material, which could destroy the bacteria cell and kill the bacteria. Furthermore, the extract has been revealed to inhibit the H+-ATPase bacterial pump, limiting the provision of energy for cellular function and proliferation of bacteria. 

3. What causes the extracts' MIC and MBC values to vary for various bacterial strains?

Response: Dear Reviewer, thanks for the question. The difference may be probably due to each pathogen's intrinsic characteristic and specific resistance feature. Moreover, the various phyto-components in each extract may have different antibacterial targets. This explanation has been added to the discussion section.

4. Given the synergy between plant extracts and antibiotics, how can Nephelium lappaceum and Hyphaene thebaica extracts boost antibiotic efficacy?

Response: Dear Reviewer, thanks for the pertinent question. The bacteria used in the study are known to over-express active efflux which is one of the main resistance strategies resulting in multiple-drug resistance in bacteria. The noteworthy synergistic interactions recorded may suggest that the extracts block the efflux machinery in bacteria thereby favoring an increased concentration of antibiotics within the bacterial cell, enough to produce inhibitory or killing effects. Moreover, extracts may alter the bacterial cell membrane, thereby increasing the cell permeability to various agents including antibiotics. Furthermore, possible interactions with key bacterial enzymes (such as beta-lactamases) can inhibit their activities thereby preventing further destruction of antibiotics having beta-lactam moiety such as imipenem used in our study. This additional explanation has been added to the discussion section.

5. What does the MBC/MIC ratio of < 4 indicate for most extracts, and how does it affect their bactericidal vs bacteriostatic properties?

Response: Dear Reviewer, thanks for the question. The extracts having MBC/MIC ratio ≤ 4 were considered to have a bactericidal effect. This suggests the ability of the specified extracts to produce drug concentrations that will kill 99.9% of the organisms exposed. If the MBC/MIC ratio is high (MBC/MIC >4), it may be impossible to safely deliver quantities of the antibiotic that kill 99.9% of the bacteria, and the drug is classified as bacteriostatic. This explanation has been added to the discussion section.

6. How does Nephelium lappaceum leaf extract's capacity to lower pH by inhibiting H⁺-ATPase relate to its antibacterial effectiveness?

Response: Dear Reviewer, thanks for the question. The proton pumps (H⁺-ATPase) in bacteria are responsible for maintaining the homeostatic media required to function correctly. The inhibition of H+-ATPase-dependent proton pumps can lead to a decrease in the survival of bacteria, as they require a certain level of energy in the medium for metabolism, growth, and multiplication. Th

---

## [Decision Letter · Decision Letter 1]

20 Nov 2024

Potential of methanol extracts of Nephelium lappaceum (Sapindaceae) and Hyphaene thebaica (Arecaceae) as adjuvants to enhance the efficacy of antibiotics against critical class priority bacteria

PONE-D-24-16310R1

Dear Dr. Seukep,

We’re pleased to inform you that your manuscript has been judged scientifically suitable for publication and will be formally accepted for publication once it meets all outstanding technical requirements.

Kind regards,

Divakar Sharma

Academic Editor

PLOS ONE

Additional Editor Comments (optional):

Accept

Reviewers' comments:

Reviewer's Responses to Questions

**Comments to the Author**

1. If the authors have adequately addressed your comments raised in a previous round of review and you feel that this manuscript is now acceptable for publication, you may indicate that here to bypass the “Comments to the Author” section, enter your conflict of interest statement in the “Confidential to Editor” section, and submit your "Accept" recommendation.

Reviewer #3: All comments have been addressed

Reviewer #4: All comments have been addressed

Reviewer #5: All comments have been addressed

2. Is the manuscript technically sound, and do the data support the conclusions?

Reviewer #3: Yes

Reviewer #4: Partly

Reviewer #5: Yes

3. Has the statistical analysis been performed appropriately and rigorously? 

Reviewer #3: Yes

Reviewer #4: Yes

Reviewer #5: Yes

4. Have the authors made all data underlying the findings in their manuscript fully available?

Reviewer #3: Yes

Reviewer #4: Yes

Reviewer #5: Yes

5. Is the manuscript presented in an intelligible fashion and written in standard English?

Reviewer #3: Yes

Reviewer #4: Yes

Reviewer #5: Yes

6. Review Comments to the Author

Reviewer #3: The manuscript has modified well by the authors and the scientific level of the manuscript is acceptable for publication.

Reviewer #4: Manuscript Number PONE-D-24-16310R1; entitled Potential of methanol extracts of Nephelium lappaceum (Sapindaceae) and Hyphaene thebaica (Arecaceae) as adjuvants to enhance the efficacy of antibiotics against critical class priority bacteria

The manuscript has improved to some extent and it could be accepted for publication.

Reviewer #5: The author addressed all questions. No further revisions needed. I recommend acceptance of this paper.

7. PLOS authors have the option to publish the peer review history of their article (what does this mean?). If published, this will include your full peer review and any attached files.

Reviewer #3: No

Reviewer #4: **Yes: **Mohammed Yosri

Reviewer #5: No

---

## [Editor Report · Acceptance letter]

22 Nov 2024

PONE-D-24-16310R1 

PLOS ONE

Dear Dr. Seukep, 

I'm pleased to inform you that your manuscript has been deemed suitable for publication in PLOS ONE. Congratulations! Your manuscript is now being handed over to our production team.

Kind regards, 

on behalf of

Dr. Divakar Sharma 

Academic Editor

PLOS ONE